



# Data assimilation of CrIS-NH₃ satellite observations for improving spatiotemporal NH₃ distributions in LOTOS-EUROS.

Shelley van der Graaf[1,2], Enrico Dammers[2], Arjo Segers[2], Richard Kranenburg[2], Martijn Schaap[2,3], Mark W. Shephard[4], Jan Willem Erisman[5]

[1]Vrije Universiteit Amsterdam, Earth and Climate Cluster, The Netherlands.
[2]TNO, Climate Air and Sustainability, Utrecht, The Netherlands.
[3]Freie Universität Berlin, Institute of Meteorology, Berlin, Germany.
[4]Environment and Climate Change Canada, Toronto, Ontario, Canada.
[5]Universiteit Leiden, Institute of Environmental Sciences, Leiden, The Netherlands.

*Correspondence to*: Shelley van der Graaf (s.c.vander.graaf@vu.nl)

**Abstract.** Atmospheric levels of ammonia (NH₃) have substantially increased during the last century, posing a hazard to both human health and environmental quality. The atmospheric budget of NH₃, however, is still highly uncertain due to an overall lack of observations. Satellite observations of atmospheric NH₃ may help us in the current observational and knowledge gaps. Recent observations of the Cross-track Infrared Sounder (CrIS) provide us with daily, global distributions of NH₃. In this

study, the CrIS-NH₃ product is assimilated into the LOTOS-EUROS chemistry transport model using two different methods aimed at improving the modelled spatio-temporal NH₃ distributions. In the first method NH₃ surface concentrations from CrIS are used to fit spatially varying NH₃ emission time factors to redistribute model input NH₃ emissions over the year. The second method uses the CrIS-NH₃ column data to adjust the NH₃ emissions using a Local Ensemble Transform Kalman Filter (LETKF) in a top-down approach. The two methods are tested separately and combined, focusing on a region in western

Europe (Germany, Belgium, and the Netherlands). In this region, the mean CrIS-NH₃ total columns were up to a factor 2 higher than the simulated NH₃ columns between 2014 and 2018, which, after assimilating the CrIS-NH₃ columns using the LETKF algorithm, led to an increase of the total NH₃ emissions of up to approximately 30%. Our results illustrate that CrIS-NH₃ observations can be used successfully to estimate spatially variable NH₃ time factors, and improve NH₃ emission distributions temporally, especially in spring (March to May). Moreover, the use of the CrIS-based NH₃ time factors resulted

in an improved comparison with the onset and duration of the NH₃ spring peak observed at observation sites at hourly resolution in the Netherlands. Assimilation of the CrIS-NH₃ columns with the LETKF algorithm is mainly advantageous for improving the spatial concentration distribution of the modelled NH₃ fields. Compared to in-situ observations, a combination of both methods led to the most significant improvements in modelled monthly NH₃ surface concentration and NH₄⁺ wet deposition fields, illustrating the usefulness of the CrIS-NH₃ products to improve the temporal representativity of the model

and better constrain the budget in agricultural areas.





## 1. Introduction

Ammonia ($NH_3$) is an alkaline gas in the Earth's atmosphere. $NH_3$ is highly reactive and readily reacts with available acids, forming aerosol components harmful to human health (Pope et al., 2009, Lelieveld et al., 2015, Giannakis et al., 2019) and, directly and indirectly, impacting global climate change (Erisman et al, 2011, Myhre et al., 2013). $NH_3$ is emitted from a large number of sources, including agriculture, natural nitrogen fixation in oceans and plants, volcanic eruptions, and biomass-, industrial- and fossil fuel burning (Erisman et al., 2015). Globally, agriculture is the largest source of $NH_3$. Agricultural emissions of $NH_3$ consist of, among others, volatilized $NH_3$ after manure and chemical fertilizer application, livestock housing and grazing and harvesting of crops. About 40% of the total global $NH_3$ emissions follow directly from volatilization of animal manure and chemical fertilizer, a spatially variable process highly controlled by the temperature and acidity of soils (Sutton et al., 2013). In western Europe, for instance, agriculture is an even more dominant source of $NH_3$ and contributes to 85-100% of all $NH_3$ emissions (Hertel et al., 2011). After the emitted $NH_3$ is transported through the atmosphere, it is deposited back to the Earth's surface through the processes of wet and dry deposition. Excess amounts of reactive nitrogen deposition can cause several adverse effects, such as eutrophication in aquatic ecosystems and soil acidification (Erisman et al., 2007) and biodiversity loss in terrestrial ecosystems (Bobbink et al., 2010).

Even though $NH_3$ at its current levels is an important threat to human health and environmental quality, its atmospheric budget is still very uncertain. $NH_3$ concentrations are highly variable in space and time and are difficult to be reliably measured in-situ due to the sticky nature of $NH_3$ leading to potential adsorption to parts of the measurement devices (von Bobrutzki et al, 2010). Globally, only a few $NH_3$ measurement networks exist, most of which contain only a small number of locations. Moreover, most measurements are performed at a coarse temporal resolution (weeks to months), while most atmospheric processes occur on much shorter time scales. Due to the lack of dense and precise measurement networks, measures for $NH_3$ emission controls currently rely mostly on estimates from models, for instance from chemical transport models (CTMs). CTMs simulate atmospheric processes such as emissions, transport, deposition and chemical conversion to estimate the spatial and temporal distribution of atmospheric $NH_3$. However, these models involve large uncertainties. On the one hand, model assumptions and parameterizations are uncertain due to insufficient or lack of knowledge of some of the processes, for instance, the limited understanding of bi-directional fluxes of $NH_3$ (Schrader and Brümmer, 2014, Schrader et al., 2018) or the direct effect of meteorology on $NH_3$ emissions (Sutton et al., 2013). On the other hand, uncertainties stem from the underlying input data and the spatial and temporal variation in emissions. Compared to other air pollutants, $NH_3$ emission inputs are especially uncertain due to their large spatiotemporal variability resulting from the diverse nature of agricultural sources (Behera et al., 2013). In Europe, the uncertainty of the total annual reported $NH_3$ emissions on a country level is for instance already estimated to be at least round ~30% (EEA, 2019). Naturally, $NH_3$ emissions from individual sources have a much higher uncertainty due to errors related to spatial and temporal redistribution. So as to reduce the uncertainty in modelled $NH_3$ fields from CTMs, it is vital to better understand both the spatiotemporal distribution of $NH_3$ emissions.



With the scarceness of in-situ measurements and uncertainties in existing models, the atmospheric $NH_3$ budget remains among the least known parts of the nitrogen cycle (Erisman et al., 2007). Recent satellite observations of $NH_3$ in the lower troposphere can help us to fill in both observational and knowledge gaps. Satellite instruments, such as the NASA Tropospheric Emission Spectrometer (TES) (Beer et al., 2008), ESA's Infrared Atmospheric Sounder Interferometers (IASI) (Clarisse et al., 2009), the NASA Atmospheric Infrared Sounder (AIRS) (Warner et al., 2016), the Thermal And Near-infrared Spectrometer for

Observation-Fourier Transform Spectrometer (TANSO-FTS) (Someya et al., 2020) and the NASA/NOAA Cross-track Infrared Sounder (CrIS) (Shephard and Cady-Pereira, 2015) provide global observations of atmospheric $NH_3$. Out of the operational satellites that observe $NH_3$ with twice daily global coverage, CrIS is the newest instrument and has the lowest radiometric noise in the spectral region used for $NH_3$ (Zavyalov et al., 2013). Moreover, CrIS has increased vertical sensitivity for near-surface $NH_3$, and provides retrievals of the vertical distribution of $NH_3$ (Shephard et al., 2020).


Measurements of atmospheric trace gases with satellites have opened up new ways to study the atmospheric budget. Recently, satellite observations have successfully been used for direct estimates of emissions and lifetimes of various other atmospheric species (e.g., $SO_2$, $NO_2$, $CO_2$) of single anthropogenic or natural point sources (e.g., Fioletov et al., 2015, Nassar et al., 2017) or even multiple sources at a time (Fioletov et al.,2017, Beirle et al., 2019). For $NH_3$ specifically, multiple studies have reported

emissions and atmospheric lifetime estimates either based on satellite data (e.g., Zhu et al., 2013, Whitburn et al., 2015, Van Damme et al., 2018, Zhang et al., 2018, Cao et al., 2020, Evangeliou et al., 2021) or directly estimated from satellite data (e.g., Van Damme et al., 2018, Adams et al., 2019, Dammers et al., 2019). Here, also different forms of model inversions have been used. Overall, these studies indicate an underestimation of both anthropogenic and natural $NH_3$ emissions in the current emission inventories. In addition to estimating $NH_3$ emissions, various studies used satellite observations to estimate dry

deposition fluxes of $NH_3$ (Kharol et al., 2018, Van der Graaf et al., 2018, Lui et al., 2020).

In this manuscript, we describe two methods to improve both the temporal and spatial variation of $NH_3$ emissions in the LOTOS-EUROS chemistry transport model with CrIS-$NH_3$ observations. The first method aims at deriving an improved set of a-priori, observation-based $NH_3$ time factors to be used for the temporal distribution of agricultural emission sources within

LOTOS-EUROS. In this method, the temporal variation of $NH_3$ surface concentrations from CrIS is used. The second method is used to assimilate the CrIS-$NH_3$ observations into the LOTOS-EUROS model. For this, a Local Ensemble Transform Kalman Filter (LETKF) approach is used as data-assimilation system, which strength lies in enhancing existing spatial patterns. The impact of the two methods, both individually and combined, on the simulated $NH_3$ emissions, concentration and deposition fields is then evaluated. The focus region of the study is a low-to-high $NH_3$ emission area within western Europe (The

Netherlands, Germany, Belgium), which is representative for other intense agricultural regions in the world. Moreover, the $NH_3$ emissions within this region are relatively well known and in-situ observations are sufficiently available.



## 2. Methodology

### 2.1. LOTOS EUROS

LOTOS-EUROS is an Eulerian chemistry transport model (Manders et al., 2017) that could be used to simulate trace gas and
aerosol concentrations in the lower troposphere. The model has an intermediate complexity with limited run time, allowing
ensemble-based simulations and assimilation studies. LOTOS-EUROS uses meteorological data as input, which in this study
is taken from the using European Centre for Medium-Range Weather Forecasts (ECMWF). The gas-phase chemistry follows
a carbon-bond mechanism (Schaap et al., 2008). The dry deposition fluxes are calculated with the Deposition of Acidifying
Compounds (DEPAC) 3.11 module, following the resistance approach and it includes a calculation of bi-directional $NH_3$ fluxes
(Van Zanten et al., 2010, Wichink Kruit et al., 2012). The wet deposition fluxes are computed using the CAMx (Comprehensive
Air quality Model with Xtensions) approach, which includes both in-cloud and below-cloud scavenging (Banzhaf et al., 2012).
The anthropogenic emissions are taken from CAMS-REG-AP (Copernicus Atmospheric Monitoring Services Regional Air
Pollutants) emissions dataset v2.2 (Granier et al., 2019). For Germany, high resolution gridded $NH_3$ emission inputs (GRETA)
are used (Schaap et al., 2018). In this study, a region in Western Europe (47°N-56°N, 2°E-16°E) is modelled, which includes
all of Germany, the Netherlands and Belgium (Fig. 2). A spatial resolution of 0.20° longitude by 0.10° latitude is used,
corresponding to approximately 12 by 12 square kilometers, which is also roughly the footprint size of CrIS (14 by 14 km$^2$ at
nadir). The vertical grid extends up to   200 hPa and is split up into 13 vertical layers. This captures the largest part of
atmospheric $NH_3$, as it is a relatively short-lived species mainly located in the boundary layer. The interfaces of the vertical
layers are based on the pressure layers used in the ECMWF meteorological input data. LOTOS-EUROS is part of the
operational Copernicus Atmosphere Monitoring Service (CAMS) ensemble forecasts and analysis for Europe (Marécal et al.,
2015). The model has participated in multiple model intercomparison studies (e.g., Bessagnet et al., 2016, Colette et al., 2017,
Vivanco et al., 2018), showing overall good performance.

### 2.2. Datasets

#### 2.2.1. CrIS $NH_3$

The Cross-Track Infrared Sounder (CrIS) is an instrument aboard NASA's sun-synchronous, Earth orbiting Suomi NPP
satellite with an equatorial overpass at 13:30 and 1:30 LST. The CrIS sensor has a spectral resolution of 0.625 cm$^{-1}$ (Shephard
et al., 2015) and a detection limit of 0.3-0.5 ppbv under favorable conditions (Shephard et al., 2020). The instrument has a
wide swath of up to 2200 km with pixels of approximately 14 km in size at nadir. Compared to other $NH_3$ satellite sounders
(e.g., AIRS, IASI), CrIS has an improved vertical sensitivity of $NH_3$ close to the surface due to its low spectral noise of
approximately 0.04K at 280K in the $NH_3$ spectral region (Zavyalov et al., 2013). Moreover, CrIS has a relatively high near-
surface sensitivity. The peak sensitivity of the instrument is typically between 900 and 700 hPa, which corresponds to
approximately 1 to 3 km (Shephard et al., 2020). The CrIS $NH_3$ total columns have an estimated total random measurement
error of around 10-15%, and an estimated random total error of ~30%. Due to the limited vertical resolution, the $NH_3$





concentrations at individual retrieval levels have a higher random measurement error of about 10-30% and a total error of ~60-

100% (Shephard et al., 2020). Version 1.3 of the CrIS-NH$_3$ product has been evaluated against in-situ Fourier Transform Infrared (FTIR) measurements (Dammers et al., 2017) showing an overall good performance and high correlations of r~0.8. In this study, we used version 1.5 of the CrIS fast physical retrieval (FPR)-NH$_3$ product, which is based on the optimal estimation method (Rodgers, 2000). More details about the CrIS FPR-NH$_3$ product can be found in (Shephard et al., 2020). Here, we used daytime observations of NH$_3$ (partial) column concentrations and surface concentrations made between January

2014 and December 2018 from the first CrIS sensor, which has the longest observing period. During this 5-year period, a virtually continuous timeseries of CrIS observations was available. More recent observations were not used due to the technical issues of the CrIS instrument during the summertime in 2019, and the potentially anomalous situation resulting from the COVID-19 outbreak in 2020.

### 2.2.2. In-situ observations

Several measurement networks were used to evaluate the simulated concentration and deposition fields. The NH$_3$ surface concentrations are evaluated against observations from the Dutch Meetnet Ammoniak in Natuurgebieden (MAN) network (Lolkema et al., 2015), the Dutch Landelijk Meetnet Luchtkwaliteit (LML) network (van Zanten et al., 2017), the Belgium Flanders Environment Agency (VMM) network (den Bril et al., 2011) and the German Environment Agency (UBA) network (Schaap et al., 2018). The locations of these sites are shown in Fig. 1. The MAN network provides monthly mean NH$_3$ surface

concentrations since 2005, spread over 80 mostly low NH$_3$ emission nature areas in the Netherlands. The measurements are performed with low-cost passive samplers from Gradko and have an estimated uncertainty of ~20% for high concentrations and ~41% for low concentrations (Lolkema et al., 2015). The NH$_3$ concentrations in Flanders are measured with passive samplers from Radiëllo and IVL samplers (den Bril et al., 2011). The LML network observes hourly NH$_3$ concentrations at six different locations in the Netherlands with different emission regimes (high, moderate, low). Initially, continuous flow

denuders from AMOR were used, which have a reported uncertainty of at least 9% for hourly concentrations (Blank et al., 2001). Around 2016, the AMOR instruments were replaced by miniDOAS instruments (Berkhout et al., 2017), which are active differential optical absorption spectroscopes. For evaluation of the wet deposition fields, observations from wet-only samplers from the Dutch Landelijk Meetnet Regenwatersamenstelling (LMRe) network (van Zanten et al., 2017), whose locations largely overlap with the LML network, and the UBA network (Schaap et al., 2018) are used. The locations of the

wet-only samplers are shown in Fig. S1.

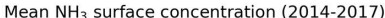

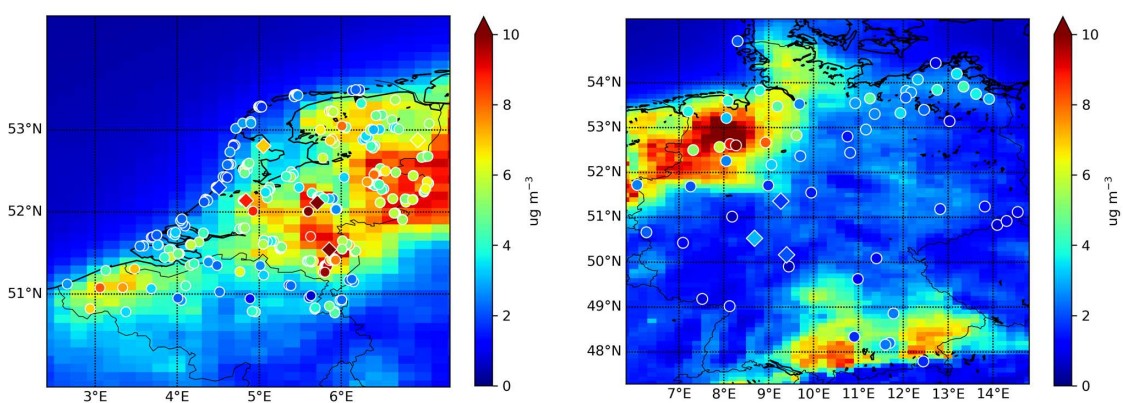

**Figure 1: Locations of stations that measure NH₃ surface concentrations. The circles depict passive samplers and the diamonds hourly observations stations.**

### 2.3. Fitting method for deriving CrIS-based NH₃ time factors

A non-linear least squares method is used to fit a trimodal gaussian curve to scaled NH₃ surface concentrations from CrIS. The Trust Region Reflective algorithm is used to perform the minimization (Conn et al., 2000). The minimalization algorithm is restrained with initial parameter guesses and bounds for three fitted gaussians. The three gaussians represent the spring ($\mu_1$, $\sigma_1$, and $A_1$), autumn ($\mu_2$, $\sigma_2$, and $A_2$) and summer peak ($\mu_3$, $\sigma_3$, and $A_3$) in NH₃ emissions, respectively. The initial parameter guesses are based on the standard MACC-III (Kuenen et al., 2014) NH₃ emission time profiles. The bounds are defined as follows:

- the mean values ($\mu_{1,2,3}$) are permitted to shift by one month (30 days) to cover the most probable emission peaks
- the standard deviations ($\sigma_{1,2,3}$) are permitted to vary by half their initial value guess (i.e., $\pm 0.5\sigma$)
- the fitted amplitude of the spring peak ($A_1$) is allowed to be between 0.1 and 1.0 and amplitudes of the autumn and summer gaussians ($A_{2,3}$) between 0.1 and 0.8

An overview of the permitted parameter bounds is given in Table 1. The range in permitted $A_{1,2,3}$ values is quite large, allowing the minimization algorithm to fit meaningful trimodal curves for different types of time variant NH₃ emission sources simultaneously (e.g., flatter peaks for emissions that mainly dependent of temperature and specific periods, such as open stables, a sharper more distinct spring and autumn peaks for emissions following fertilizer or manure application).


| | Spring peak | | | Autumn peak | | | Summer peak | | |
|---|---|---|---|---|---|---|---|---|---|
| | $\mu_1$ (doty) | $\sigma_1$ (days) | $A_1$ (-) | $\mu_2$ (doty) | $\sigma_2$ (days) | $A_2$ (-) | $\mu_3$ (doty) | $\sigma_3$ (days) | $A_3$ (-) |
| Lower bound | 47.4 | 13.1 | 0.1 | 222.8 | 11.6 | 0.1 | 148.9 | 26.9 | 0.1 |
| First guess (MACC-III) | 77.4 | 26.1 | 0.96 | 252.8 | 23.2 | 0.26 | 178.9 | 53.7 | 0.21 |
| Upper bound | 107.4 | 39.1 | 1.0 | 282.8 | 34.8 | 0.8 | 208.9 | 107.4 | 0.8 |

Table 1: Initial parameter guesses and parameter bounds used in the trimodal fit algorithm.

After the daily $NH_3$ time factors are fitted, the diurnal variation from the MACC-III $NH_3$ time factors is added to obtain hourly time factors. The resulting hourly CrIS-based time factors are used as input for all time-variant $NH_3$ sources from agriculture subcategories in LOTOS-EUROS, i.e., continuous $NH_3$ point sources emissions remain time-invariant.

**2.3.2. Data selection**

The CrIS $NH_3$ concentrations in the lowest retrieval level, i.e., closest to the surface, are used to adjust the hourly time profiles
spatially on a regular 0.1° by 0.05° grid. First, to collect enough observations, the CrIS $NH_3$ surface concentrations with a quality flag of at least 3 and within a selection radius of 1° around the center points of each grid cell are selected. The daily average $NH_3$ concentrations throughout the year are computed after application of a simple outlier filter (>99[th] percentile excluded given more than 3 observations). Due to the lower number of observations during winter, and to avoid a bias towards higher values due to lower thermal contrast, observations in January, November and December are ignored. During these
months it is anyway prohibited to apply fertilizer or spread manure in parts of the regions (for the Netherlands, see RVO, 2021), and in combination with the colder temperatures, $NH_3$ concentrations are expected to be low due to low volatilization rates (e.g., Søgaard et al., 2002).

**2.3.3. Correction for local emission to concentration ratio**

The relationship between $NH_3$ emissions and surface concentrations differs per region and changes throughout the year due to
changes in the meteorological and chemical circumstances. To correct for this, the following adjustment factor is applied to the daily CrIS $NH_3$ surface concentrations. The factor is based on the $NH_3$ emission and simulated surface concentration fields from LOTOS-EUROS, which are used to compute the local ratio of the smoothed daily total $NH_3$ emissions to the $NH_3$ surface concentrations at the CrIS overpass time per grid cell. These are used as a first order approximation for the relation between the emission and concentration. The ratios are rescaled by the mean annual values for each grid cell to obtain a unitless daily
scaling factor (Fig. S2). After multiplying the daily averaged CrIS $NH_3$ surface concentrations with this scaling factor, a ±1σ filter is used to smoothen out the daily time series. To avoid too much flattening of the spring emission peak, a separate filter is applied for the spring period. Finally, the scaled $NH_3$ surface concentrations are normalized for each grid cell.



### 2.4. Data assimilation system

### 2.4.1. Local Ensemble Transform Kalman Filter

The Ensemble Kalman Filter (Evensen, 2003) is a sequential data assimilation method that could be used to combine model simulations with observation. In this study, the Local Ensemble Transform Kalman Filter (LETKF) formulation is used (Hunt et al, 2007) following the implementation by (Shin et al., 2016). The LETKF performs an analysis per grid cell based on nearby observations only and it therefore computationally advantageous compared to the regular implementation of the Ensemble Transform Kalman Filter. The basic idea behind an Ensemble Kalman Filter is to express the probability function of the state

in terms of an ensemble with N possible states $x_1$, $x_2$, … $x_N$, each considered to be a possible sample out of the distribution of the true state. In this study, the state contains the $NH_3$ concentrations in a three-dimensional grid and two-dimensional $NH_3$ emission perturbation factors β. The perturbation factors describe the uncertainty in the emissions, and are modelled as samples out of normal distribution with zero mean and standard deviation σ. Spatial variations are initially not defined, but are introduced by a localization length scale that is described below. The temporal variation in the emission factors is described

by temporal correlation coefficient α, that depends on temporal length scale τ (Lopez-Restrepo et al., 2020, Barbu et al., 2009):

$$\alpha_k = e^{-|t_k - t_{k-1}|/\tau} \tag{Eq. 1}$$

An initial ensemble is created by generating random samples of the perturbation factors. The ensemble is then propagated in time in what is called the *forecast* step between consecutive *analysis* times for which observations are available. In the *forecast*

step, the model propagates the *analysed* ensemble members from time $t_{k-1}$ to time $t_k$ following:

$$\mathbf{x_i}(k) = \mathbf{M_{k-1}}(\mathbf{x_i^a}(k-1)) \tag{Eq. 2}$$

where operator $\mathbf{M_{k-1}}$ describes the model simulation, including application of the perturbation factors that are present in $\mathbf{x}$. The ensemble mean $x$ and forecast error covariance $\mathbf{P}$ at time $k$ are expressed as:

$$x = \frac{1}{N}\sum_{i=1}^{N} x_i \tag{Eq. 3}$$

$$\mathbf{P} = \frac{1}{N-1}\sum_{i=1}^{N}(x_i - x)(x_i - x)^T \tag{Eq. 4}$$

When CrIS observations ($\mathbf{y^{CrIS}}$) are available (at time $t_k$), the LETKF algorithm *analyses* the ensemble by incorporating the observations to reduce the ensemble spread. The *analysed* ensemble members are computed from:

$$\mathbf{x_i^a} = \mathbf{x_i} + \mathbf{P^a H^T R^{-1}}(\mathbf{y^{CrIS}} - \mathbf{h}(x_i) + \mathbf{v_i}) \tag{Eq. 5}$$

In here, $\mathbf{h}(x_i)$ represents the simulated retrieval from the state $\mathbf{x_i}$, or in particular from the concentration array in $\mathbf{x_i}$. Operator $\mathbf{H}$ is a linearization of $\mathbf{h}(x)$ to $x$. The matrix $\mathbf{R}$ is the *observation representation error covariance*, which describes the





difference between the simulation and the observation due to measurement and representation errors:

$\mathbf{y^{CrIS}} - \mathbf{h}(x_i) \quad \sim N(0, \mathbf{R})$ (Eq. 6)

The actual implementation of $\mathbf{h}$, $\mathbf{H}$, and $\mathbf{R}$ are described below. The *analysis covariance* $\mathbf{P^a}$ is computed from:

$\mathbf{P^a} = [\mathbf{PH^TR^{-1}H + I}]^{-1}\,\mathbf{P}$ (Eq. 7)

### 2.4.2. Observation simulation

The simulated observation vector $\mathbf{h}(x_i)$, representing the simulated retrieval, is computed from:

$\mathbf{h}(x_i) = \mathbf{y_a} - \mathbf{Ay_a} + \mathbf{AG}x_i$ (Eq. 8)

Here, matrix $\mathbf{G}$ is applied to horizontally and vertically match the simulated partial $NH_3$ columns in LOTOS-EUROS with the retrieval CrIS pressure levels. The relationship between the true and the retrieved atmospheric $NH_3$ profiles, i.e., the vertical

sensitivity of the CrIS measurements, is described by averaging kernel $\mathbf{A}$. The full relationship between the true and the observed state is given by (Eq. 8) (Rodgers and Connor, 2003):

$\mathbf{y^{true}} = \mathbf{y_a} + \mathbf{A}\,(\mathbf{G}\,\mathbf{x^{true}} - \mathbf{y_a}) + \mathbf{v}$ (Eq. 9)

with $\mathbf{y_a}$ the a-priori profile that is part of the CrIS retrieval product. The error $\mathbf{v}$ is a sample of the *observation representation*

*error* that describes the possible differences between simulation and retrieval:

$\mathbf{v} \sim N(0, \mathbf{R})$ (Eq. 10)

In this study, $\mathbf{R}$ is set to the retrieval error covariance that is part of the CrIS product. The linearized observation operator becomes:

$\mathbf{H} = \mathbf{A}\,\mathbf{G}$ (Eq. 11)

### 2.4.3. Analysis per grid cell

The analysis described above is applied per model grid cell; for the exact implementation we refer to Shin et al. (2016). First, the simulated observation vectors $\mathbf{h}(x_i)$ are computed for all ensemble members. For the grid cell to be analyzed, all simulations are collected that are within $3.5\rho$ distance, where $\rho$ is called the localization length scale as well as the matching

actual observations $\mathbf{y^{CrIS}}$. The state elements corresponding to the grid cell are then analyzed using the collected observations and simulations, where the weight of observations further away is limited using Gaussian correlation that decays with distance and that uses the same correlation length scale $\rho$ that is used for collection.



### 2.4.4. Observation selection

CrIS observations with the highest quality flag, QF = 5, were used. As the assumed vertical NH₃ profile shape in background areas used in the CrIS retrieval and in LOTOS-EUROS differ, CrIS retrievals with "unpolluted" a-priori profiles were filtered out. The original CrIS retrieval is performed in the log domain and therefore either the averaging kernels A from CrIS need to be linearized or the LOTOS-EUROS profiles transformed to the log-domain. Linearization of the kernel is only accurate for higher concentrations, and since this is the case for the selected "polluted" retrievals, this option was found to be suitable.

### 2.4.5. Parameter calibration

In this study, we used a localization radius of $\rho = 15$ km, a standard deviation of $\sigma = 0.5$ and a temporal correlation length of $\tau = 3$ days. Two experiments were performed to study the effect of $\rho$, $\sigma$ and $\tau$ in more detail. A description of the experiments can be found in section S1 of the supplementary materials. A limited ensemble size of N=12 was found to be sufficient to describe the imposed model uncertainty, which is not too complicated due to short life-time of NH₃ and therefore strong relation between concentrations and nearby emissions.

## 3. Results

### 3.1. Direct comparison of NH₃ concentrations from CrIS and LOTOS-EUROS

Before looking at the effects of assimilating the CrIS observations, a direct comparison of the modelled and observed NH₃ column densities was made. The simulated NH₃ concentrations from the default run in LOTOS-EUROS were sampled at the locations of the CrIS observations, and after application of the averaging kernels compared with the retrievals. The observed and simulated NH₃ total columns averaged over all years are shown in Fig. 2. Similar maps per year are available in Fig. S3 of the supplementary materials. The general pattern of the NH₃ total column densities matches quite well. For instance, the observed and simulated NH₃ columns are very similar in southwestern Germany, and close to the Dutch border. The CrIS NH₃ total columns are generally higher than the simulated NH₃ total columns. This is for instance found in large parts of northeastern Germany, along the Belgium coast and in the south of the Netherlands. Here, the observed NH₃ columns were on average approximately a factor 2 higher than the simulated NH₃ columns. Moreover, the observed NH₃ total columns are consistently higher than the simulated NH₃ columns in background areas, with a bias between the observed and modelled concentrations of approximately ~0.5x10¹⁶ molecules/cm².



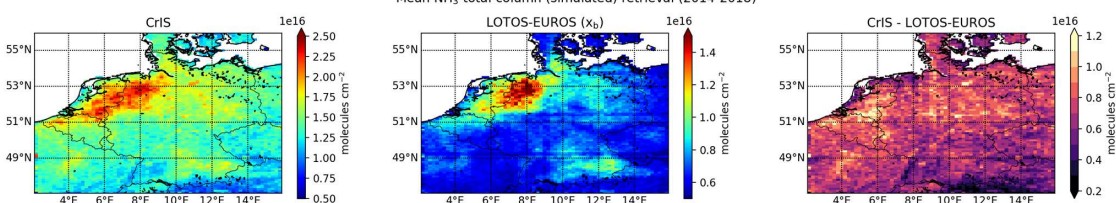

**Figure 2: Mean retrieved (left) and simulated (middle) NH₃ total column from 2014-2018, and their absolute difference (right).**

### 3.2. CrIS-based NH₃ time factors

#### 3.2.1. Effect on NH₃ emissions in LOTOS-EUROS

Following the method described in section 2.3, temporal profiles for the NH₃ have been obtained per grid cell. Compared to the original model, the new time profiles vary spatially. Fig. 3 shows a comparison of the daily grid-averaged NH₃ emissions between the default background model run ($x_b$) and the background run with the CrIS-based NH₃ time factors ($x_{b,CrIS}$), using a different color for each month. The default NH₃ time factors from MACC-III provide more intra-annual variation than the CrIS-based NH₃ time factors. The default time factors include a very high peak in spring and much lower peaks during summer and autumn (i.e., $A_1/A_3 = 4.57$, $A_1/A_2 = 3.70$). Fig. S4 shows the fitted spring parameters ($\mu_1$, $\sigma_1$ and $A_1$). The NH₃ spring peak present in the CrIS-NH₃ surface concentrations is generally lower than the default NH₃ spring peak. In large parts of the model region, the CrIS-observed NH₃ spring peak is subsequently lower and less sharp. Compared to the default NH₃ time factors, the amplitude of the spring peak in the CrIS-based NH₃ time factors is now generally much lower. The amplitude of the spring peak differs almost by a factor 2 on average. As a result, there is a decrease in springtime NH₃ emissions, especially in March and April. The CrIS-based NH₃ time factors, and consequently the NH₃ emissions, are, on the other hand, generally higher later in the year. The NH₃ emissions are on average approximately 50% higher in summer and the beginning of autumn (June to September), and approximately twice as high in October.





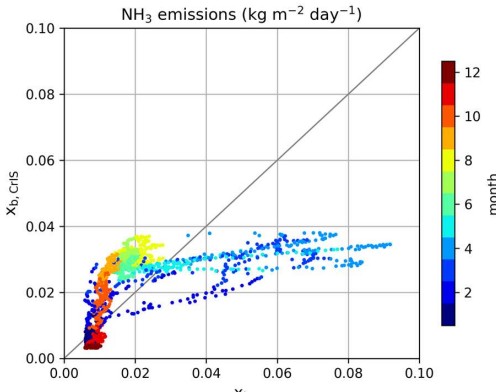


**Figure 3: Daily grid-average NH₃ emission, colored per month. Here, $x_b$ represents the default background run and $x_{b,CrIS}$ the background run with CrIS-based NH₃ time factors.**

### 3.2.2. Effect on NH₃ concentrations and deposition fields in LOTOS-EUROS

The changes in modelled NH₃ surface concentration, total column concentrations and $NH_x$ total deposition from 2014 to 2018

related to the use of the CrIS-based NH₃ time factors are shown in Fig. 4, Fig. S5 in the supplementary materials and Fig. 5. Here, $x_b$ represents the default background run and $x_{b,CrIS}$ the background run with the CrIS-based NH₃ time factors. The use of the CrIS-based emission time profiles led to an overall increase in mean NH₃ surface concentrations. The absolute change is largest in areas with already relatively high NH₃ surface concentrations, for instance in northwestern Germany, where the mean NH₃ surface concentrations increased with up to 2 μg/m³. The mean NH₃ surface concentrations increased with up to

~25% due to the change in NH₃ time factors. The effect of the CrIS-based NH₃ time factors on the NH₃ total column concentrations is smaller, with minor changes from minus ~5% up to 5%. The mean NH₃ total column concentrations generally increase in areas with already high NH₃ concentrations, such as large parts of the Netherlands, and decrease in background areas with little NH₃ emissions, for instance in central Germany. The use of the CrIS-based NH₃ time factors led to ~10% less total $NH_X$ deposition along the northwestern coast, including agricultural hotspots such as the Netherlands and northwestern

Germany, and an increase of up to ~10% in background areas.

Fig. S6 compares the daily, grid averaged, NH₃ surface concentrations, total column concentrations and $NH_x$ wet and dry deposition, with different colors per month. Here, a similar redistribution is observed for the NH₃ concentration and deposition fields as seen earlier for the NH₃ emission fields. Compared to the default background run ($x_b$), the NH₃ concentration fields were up to a factor 2 lower during March and April. The NH₃ total columns decreased in spring, the largest decrease occurring

in April (up to ~60%). The NH₃ surface concentrations increased during the summer and the beginning of autumn, up to ~50% in September. During these months, a similar but slightly lower increase in the NH₃ total column concentrations is observed.





Because the CrIS-based $NH_3$ time factors vary per year, the interannual variation in the modelled $NH_3$ fields is much larger. Fig. S7 shows the relative changes in $NH_3$ surface concentration, total column concentration and $NH_x$ deposition fields per year. Overall, the mean $NH_3$ surface concentration increases by up to ~30% per year. The largest increases occurred in 2016

and 2018, years with relatively high summer temperatures (Copernicus Climate Change Service, 2021). The variation in the annual mean $NH_3$ total column concentrations is much smaller (-15 to +15%). The relative change in $NH_x$ budget shows much more variation, with the most prominent increase occurring in 2014 (+25%) and decreases occurring in 2018 (-25%).

The temporal redistribution of the $NH_3$ emissions thus significantly impacts the modelled $NH_3$ concentration and deposition fields, too. Generally, a part of the initial spring $NH_3$ emissions is now attributed to the summer and autumn months. Depending

on the degree of redistribution, there are distinct changes in the $NH_x$ budget. Looking at the fitted spring peak parameters (Fig. S4) and the matching CrIS-based $NH_3$ factors at hourly measurement sites (Fig. S8), clear interannual differences are observed. For instance, a relatively sharp spring peak was observed over the Netherlands in 2014. In 2018, on the other hand, the fitted spring peak had a distinctly lower amplitude and started later in the year. Moreover, a relatively large rise in $NH_3$ time factors was observed again in late summer and autumn (July to September). Compared to 2014, this resulted in a relatively larger

redistribution of the $NH_3$ emissions towards warmer months. The higher temperatures resulted in lower dry deposition velocities and more vertical mixing and transport of $NH_3$, leading to an overall decrease in $NH_x$ deposition over the Netherlands. Moreover, the summer of 2018 was relatively dry, also leading to higher $NH_3$ total column concentrations and a decrease in wet $NH_x$ deposition.

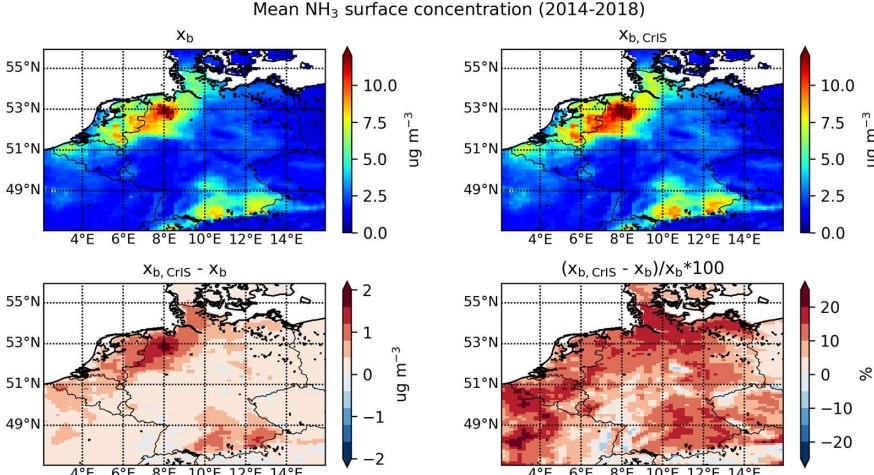

**Figure 4: The mean $NH_3$ surface concentration over 2014 to 2018 from the (top left) default background run ($x_b$) and the (top right) background run with CrIS-based $NH_3$ time factors ($x_{b,CrIS}$) and their (bottom left) absolute and (bottom right) relative difference.**

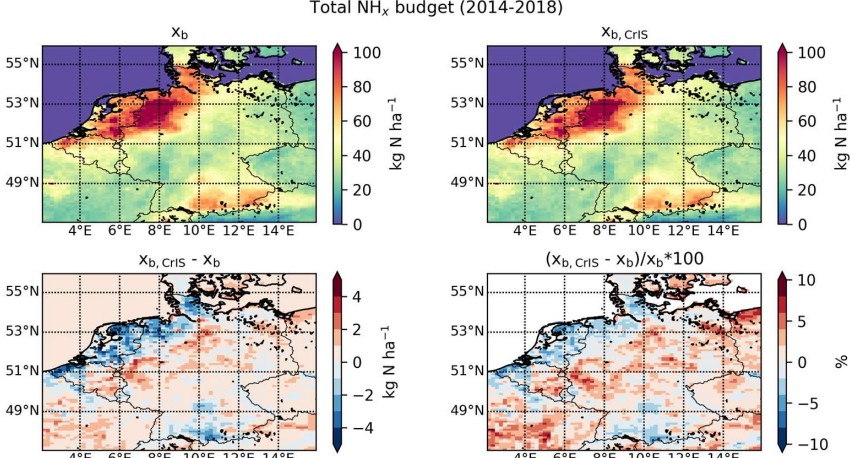

**Figure 5: The total NH$_x$ deposition from 2014 to 2018 from the (top left) default background run (x$_b$) and the (top right) background run with CrIS-based NH$_3$ time factors (x$_{b,CrIS}$) and their (bottom left) absolute and (bottom right) relative difference.**

**3.3. Local Ensemble Transform Kalman Filter**

**3.3.1. Effect on NH$_3$ emissions and concentrations in LOTOS-EUROS**

The CrIS-NH$_3$ columns were assimilated using the Local Ensemble Transform Kalman Filter (LETKF) described in section 2.4. Assimilations were performed using either the default emission time profiles (x$_a$), or using the CrIS-based profiles (x$_{a,CrIS}$). The total NH$_3$ emissions from 2014 to 2018 and the relative and absolute changes compared to background simulations x$_b$ and

x$_{b,CrIS}$ are shown in Fig. 6. The corresponding mean NH$_3$ surface and total column concentrations are shown in Fig. S9 and Fig. S10 of the supplementary materials. The absolute NH$_3$ emission updates by the LETKF are, as expected, largest in regions with already high NH$_3$ emissions. There is a maximum increase of ~30% in total NH$_3$ emission by the LETKF over the entire period for some grid cells. Relatively, the largest changes are found in the southern parts of the Netherlands (province of Noord-Brabant), the west coast of Belgium (province of West-Vlaanderen), the northeastern parts of Germany and France.

Compared to the analysis run using default emission time profiles (x$_a$), the analysis runs with the CrIS-based NH$_3$ profiles (x$_{a,CrIS}$) generally have more NH$_3$ emission and consequently higher NH$_3$ surface and total column concentrations. The long-term spatial patterns of the emission updates by the LETKF, however, remained very similar.



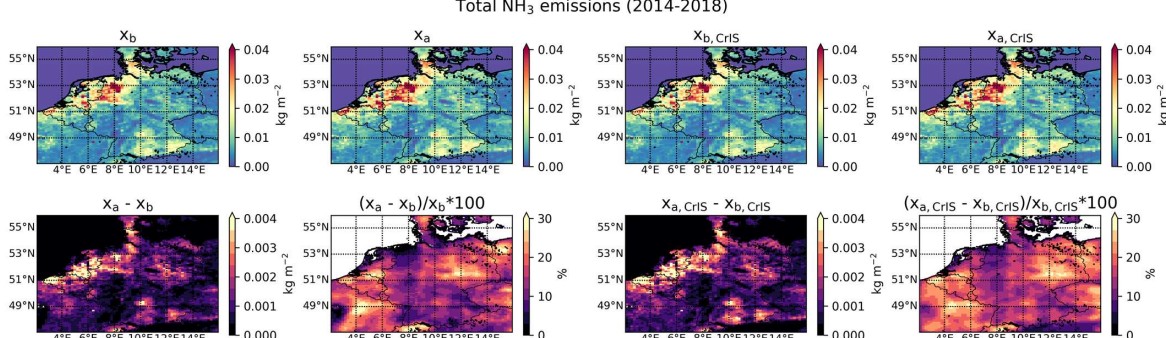

**Figure 6: The total NH$_3$ emissions in 2014-2018 in the background runs x$_b$ and x$_{b,\,CrIS}$ and in analysis runs x$_a$ and x$_{a,CrIS}$ (top panels), as well as their absolute and relative difference (bottom panels).**

To study the effect of the LETKF in more detail, the daily grid average NH$_3$ emissions of the background runs (x$_b$ and x$_{b,CrIS}$) are plotted against analysis runs (x$_a$ and x$_{a,CrIS}$) in Fig. 7. Similar for the NH$_3$ surface and total column concentrations are plotted in Fig. S11 of the supplement. In the runs with the default NH$_3$ time factors (x$_b$ and x$_a$), data assimilation of the CrIS-NH$_3$ columns led to both positive and negative emission updates in spring. In the summer, on the contrary, it mostly resulted in an increase in NH$_3$ emissions. In the runs with the CrIS-based NH$_3$ time factors (x$_{b,CrIS}$ and x$_{a,CrIS}$), the pattern is distinctly different. Compared to the default runs, the NH$_3$ emission updates in spring are now smaller and largely positive, with the largest updates occurring in April. Moreover, the NH$_3$ emission updates were generally smaller during summer, too. This is related to the fact that the CrIS-NH$_3$ surface concentrations were used to fit the NH$_3$ time factors, which resulted in the model being closer to the CrIS observations already.

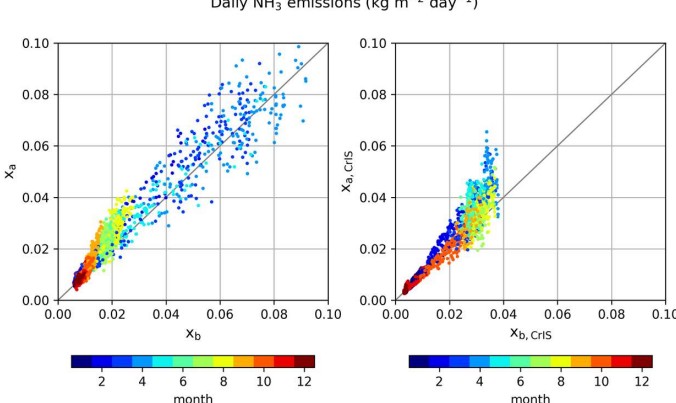

**Figure 7: Daily grid average NH$_3$ emissions in 2014-2018 from the (left) default background run (x$_b$) versus analysis run (x$_a$), and from the (right) background run with the CrIS-based NH$_3$ time factors (x$_{b,\,CrIS}$) versus analysis run x$_{a,CrIS}$, colored per month.**



Perturbation factor β is the computed multiplication factor by which the initial input $NH_3$ emissions are updated in the LETKF.
The mean perturbation factors β per year are shown in Fig. S12 of the supplementary materials. The pattern of the $NH_3$ emission updates does not change drastically between years, which points to a consistent, spatial misdistribution of the emissions in the current inventory. By far the largest mean $NH_3$ emission updates took place in 2018, followed by 2015.

Fig. 8 shows timeseries of the daily grid average $NH_3$ emissions in both background runs $x_b$ and $x_{b,CrIS}$ and analysis runs $x_a$ and $x_{a,CrIS}$. Fig. 9 and S13 show the corresponding timeseries and changes in $NH_3$ surface and total column concentrations. The
$NH_3$ emissions in the default background run ($x_b$) have a strong, annually reoccurring spring peak. After this peak, the $NH_3$ emissions decrease steeply and then slightly increase again in late summer and autumn (August and September). In analysis run $x_a$, the spring $NH_3$ emissions are both positively and negatively adjusted. Later in the year, almost only positive emission updates are found. The largest positive $NH_3$ emission updates occurred around August and September, which suggests an underestimation of the autumn $NH_3$ peak in the default runs.

In the background runs with the CrIS-based $NH_3$ time factors ($x_{b,CrIS}$), the $NH_3$ emissions are much more evenly distributed over the year. In contrast to the default runs, practically only positive $NH_3$ emission updates occurred in the analysis run ($x_{a,CrIS}$). The largest $NH_3$ updates took place during spring (March to May). The flattening of the $NH_3$ emissions led to a flattening in $NH_3$ concentration fields, too. Compared to default runs ($x_b$ and $x_a$), there is much less interannual variation in the $NH_3$ surface and total column concentrations. As a result, the $NH_3$ concentrations during summer and autumn could be at the
same level or even higher than the springtime concentrations. During the warm summer of 2018 (Copernicus Climate Change Service, 2021), for instance, the $NH_3$ concentrations in August even clearly exceed the spring $NH_3$ concentrations.

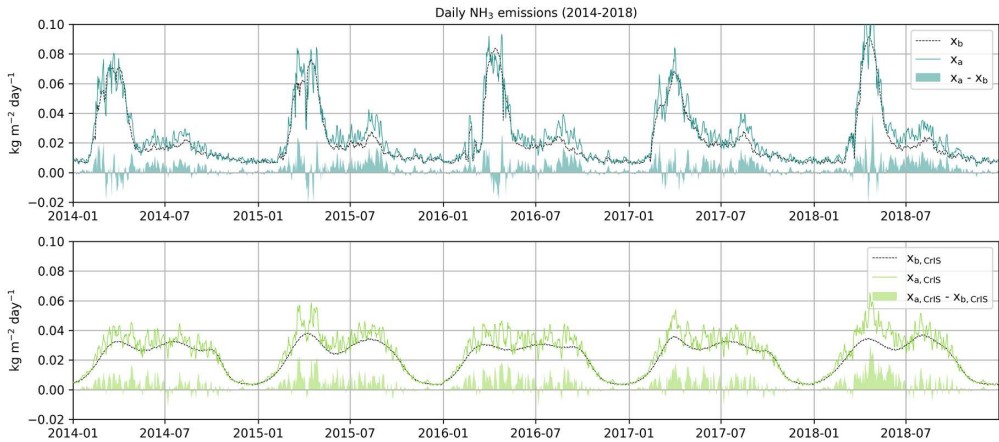

**Figure 8: Timeseries of the daily grid-averaged $NH_3$ emissions in the background and analysis runs, and their absolute difference.**
**The top figure (blue) represents the default background ($x_b$) and analysis run ($x_a$). The bottom figure (green) the background ($x_{b,CrIS}$) and analysis run ($x_{a,CrIS}$) with the CrIS-based $NH_3$ time factors.**





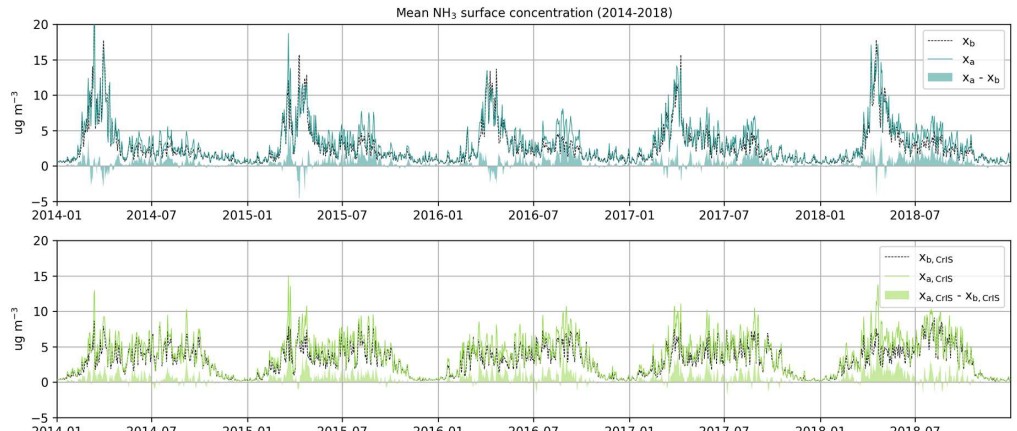

**Figure 9: Timeseries of the daily grid-averaged NH₃ surface concentrations in the background and analysis runs, and their absolute**

**difference. The top figure (blue) represents the default background (x_b) and analysis run (x_a). The bottom figure (green) the background (x_{b,CrIS}) and analysis run (x_{a,CrIS}) with the CrIS-based NH₃ time factors.**

### 3.3.2. Effect on NH_x deposition in LOTOS-EUROS

The modelled total NH_x budgets from 2014 to 2018 from the two background runs (x_b and x_{b,CrIS}) and analysis runs (x_a and x_{a,CrIS}) are shown in Fig. 10. Overall, the modelled NH_X budget shows the same spatial pattern as the NH₃ emissions. Like the

NH₃ emissions, the relatively largest spatial differences between the background and analysis runs took place in the south of the Netherlands, the west of Belgium and northeast Germany. Compared to the default runs, the relative changes in total NH_x budget were slightly larger in the runs with the CrIS-based NH₃ time factors (x_{b,CrIS} and x_{a,CrIS}).

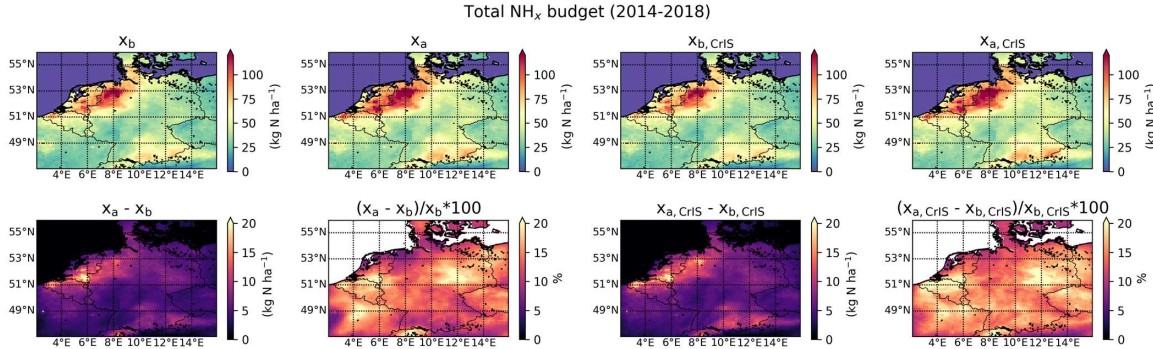

**Figure 10: The total NH_x budget from 2014-2018 in the background (x_b and x_{b, CrIS}) and analysis (x_a and x_{a,CrIS}) model runs in**

**LOTOS-EUROS, and their absolute and relative difference.**


The modelled $NH_x$ deposition follows the temporal distribution of the $NH_3$ emissions, too. Timeseries of the daily wet and dry deposition amounts in the domain are shown in Fig. 11. The wet and dry deposition in the default runs ($x_b$ and $x_{b,CrIS}$) versus the analysis runs ($x_a$ and $x_{a,CrIS}$) per month is shown in Fig. S14 in the supplement. In the default background run ($x_b$), the total $NH_x$ deposition peaks in March and April. In the analysis run ($x_a$), the dry and wet deposition both increased and decreased

during spring (March to May). Later in the year, the wet and dry $NH_x$ deposition mostly increased in the analysis run, particularly in August and September. In the background runs with the CrIS-based $NH_3$ time factors ($x_{b,CrIS}$ and $x_{a,CrIS}$), the modelled dry and wet deposition fields are much less variable. Following the $NH_3$ emission updates, both the dry and wet deposition mostly increased in the analysis run, especially in March and April. Moreover, the use of the CrIS-based $NH_3$ time factors resulted in a redistribution of the ratio of wet and dry deposition over the year. As a result of the relatively lower spring

$NH_3$ surface concentrations, there is a reduction of the dry deposition during spring. The relatively higher summer $NH_3$ total column concentrations led to a shift in wet deposition, too, from spring to summer.

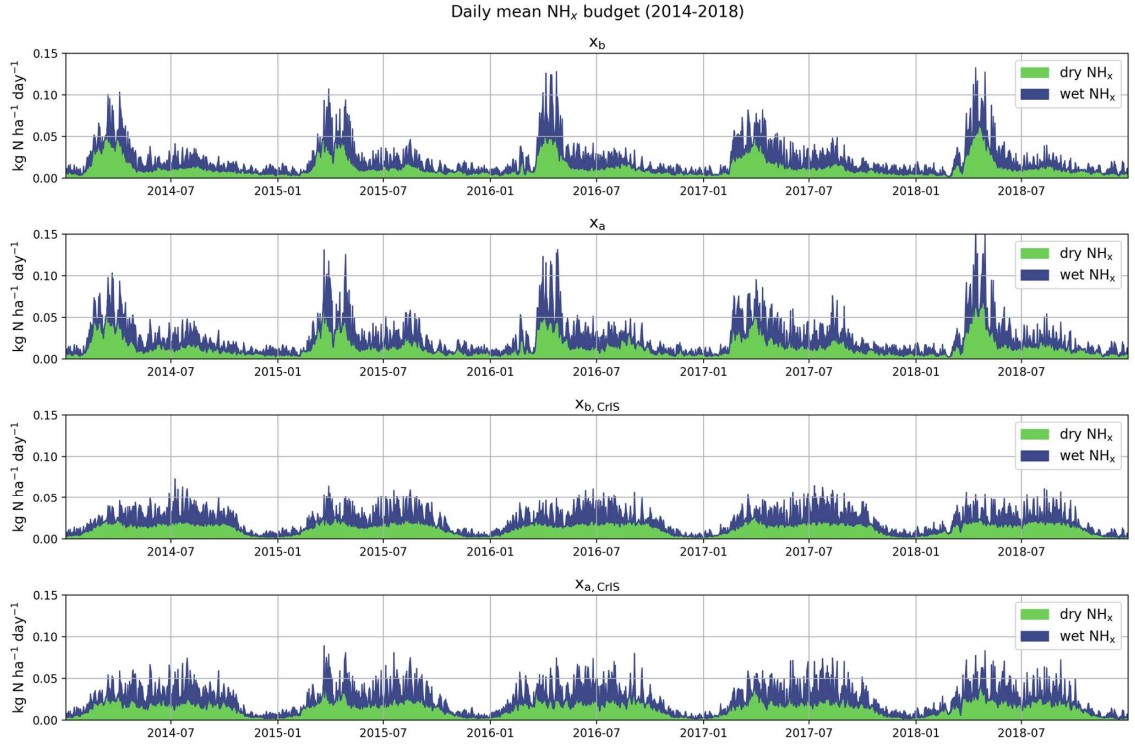

Figure 11: Timeseries of the average amounts of dry (green) and wet (blue) $NH_x$ deposition in the different model runs. The top two figures represent the default background ($x_b$) and analysis ($x_a$) run and the bottom two figures the background ($x_{b,CrIS}$) and analysis

($x_{a,CrIS}$) run with the CrIS-based $NH_3$ time factors.





### 3.4. Comparison to in-situ observations

The modelled NH$_3$ surface concentration and NH$_4^+$ wet deposition fields are compared with in-situ observations. First, the spatial distribution is evaluated by comparing the modelled NH$_3$ surface concentration and NH$_4^+$ wet deposition fields to the observed annual averages per measurement site. Second, the temporal distribution is evaluated by comparing the modelled

NH$_3$ surface concentration and NH$_4^+$ wet deposition fields to the same set of observations, but on a monthly basis. The comparisons are done per type of observation, e.g., all available wet-only measurements simultaneously. To differentiate between different NH$_3$ emission regimes, the results are plotted separately for either all hourly observations or for the passive samplers. The results are shown in Fig. 12 and 13. The Dutch site with the highest NH$_3$ surface concentrations, Vredepeel, is excluded from this comparison because of the large model-observation discrepancies here (see Fig. S18). This site is located

near agricultural emission sources and therefore less representative of a larger region. In these figures, the first column shows the comparison for the default background run (x$_b$), the second column shows the background run with CrIS-based NH$_3$ time factors (x$_{b,CrIS}$), the third column shows the analysis run with the default NH$_3$ time factors (x$_a$) and, finally, the fourth column shows the analysis run with CrIS-based NH$_3$ time factors (x$_{a,CrIS}$).

### 3.4.1. Spatial distribution

Fig. 12 shows the scatterplots of the annual averages per site per year. The annual average NH$_3$ surface concentrations (top row) in the default run x$_b$ show a strong correlation (r = 0.88) with the observed concentrations at the hourly observation sites (LML and UBA). Here, the NH$_3$ surface concentrations are generally underestimated (slope = 0.61). The annual average NH$_3$ surface concentrations (middle row) at the passive sampler sites (MAN, VVM and UBA) are generally overestimated (slope = 1.17), with a lower, but still relatively strong correlation is observed (r = 0.69). The modelled annual average NH$_4^+$ wet

deposition budgets (bottom row) are moderately correlated with the observations from wet-only samplers (r = 0.45), and are generally lower than the observed wet deposition (slope = 0.81). When using the CrIS-based NH$_3$ time factors, the annual average NH$_3$ surface concentrations and NH$_4^+$ wet deposition budgets are slightly increased. This led to a slight, overall increase in slope between all observations and the background run with the CrIS-based NH$_3$ time factors (x$_{b,CrIS}$). As the annual totals, and herewith the spatial distribution of the NH$_3$ emissions, remained the same in this run, the other measures (r, RMSE,

MAD, MRD, NMB) didn't change drastically on a yearly basis.

The comparison with annual average NH$_3$ surface concentrations from the passive sampler networks from both analysis runs (x$_a$ and x$_{a,CrIS}$) slightly worsened compared to the background runs (x$_b$ and x$_{b,CrIS}$). The comparison at the hourly observation and wet-only sampler sites, on the other hand, showed clear improvements. Here, virtually all statistical measures improved,

illustrating an overall improvement in modelled NH$_3$ surface concentration and NH$_4^+$ wet deposition field spatially. Of all runs, the analysis run with the CrIS-based NH$_3$ time factors (x$_{a,CrIS}$) compared the best with the hourly observation and wet-only sampler network. The differences between the modelled and observed NH$_3$ surface concentrations at the hourly observation





were distinctly smaller, compared to the default background run ($x_b$: {RMSE = 2.79, MAD = 1.96, MRD = -0.15, NMB = -0.28} versus $x_{a,CrIS}$: {RMSE = 2.2, MAD = 1.69, MRD = -0.11, NMB = -0.08}). Here, also the slope largely improved ($x_b$:

slope = 0.61 versus $x_{a,CrIS}$: slope = 0.76). The same is observed for the modelled NH4 wet deposition fields, where the slope improved particularly ($x_b$: {RMSE = 0.95, MAD = 0.63, MRD = -0.13, NMB = -0.22, slope = 0.81} versus $x_{a,CrIS}$: {RMSE = 0.92, MAD = 0.61, MRD = -0.02, NMB = -0.11, slope = 0.95}).

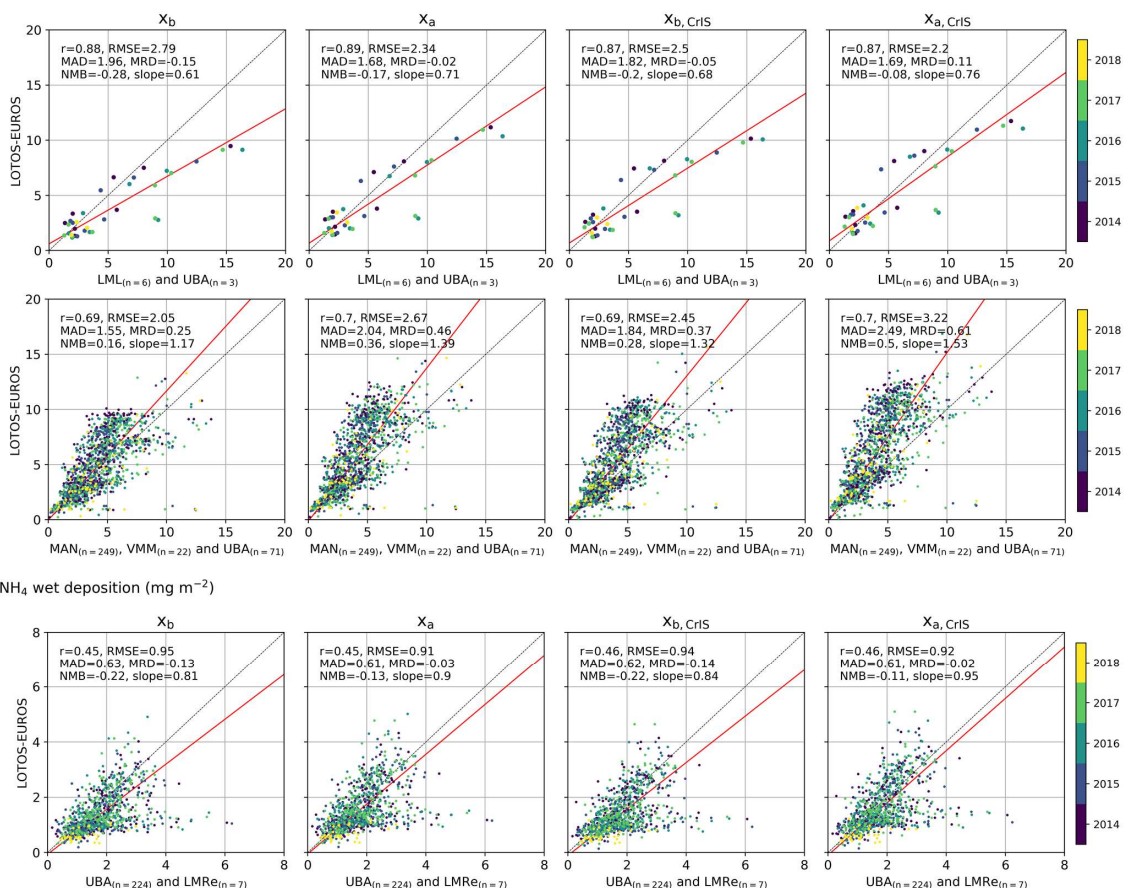

**Figure 12: Comparison of the modelled annual average NH3 surface concentrations and NH4+ wet deposition fields to**
**in-situ observations.**


### 3.4.2. Temporal distribution

Fig. 13 shows the scatterplots of the monthly means per site. The modelled monthly $NH_3$ surface concentrations from the default background run ($x_b$) are strongly correlated with the hourly observation network (r = 0.73), and with the passive sampler network (r = 0.63). Both comparisons show a distinct overestimation of the $NH_3$ surface concentration in March and April.

The observed $NH_3$ surface concentrations at the hourly observation sites are higher than the modelled ones during the rest of the year. At the passive sampler sites, the observed versus modelled monthly $NH_3$ surface concentrations during the rest of the year lie more around the one-on-one line. Here, too, the modelled $NH_3$ surface concentrations are slightly underestimated at the beginning of summer (June and July). The $NH_4^+$ wet deposition is moderately correlated with monthly observations from wet-only samplers (r = 0.44). At these sites, a similar pattern is observed. The modelled $NH_4^+$ wet deposition is overestimated

in spring (especially March and April), and underestimated during the rest of the year. In general, this comparison indicates an overestimation of the $NH_3$ spring peak emissions in the default model runs, particularly in March and April, and an underestimation of the $NH_3$ emission during the rest of the year, mainly during summer (June, July, August).

The use of the CrIS-based $NH_3$ time factors ($x_{b,CrIS}$) led to an overall improvement at the hourly observation and wet-only

sampler sites. Compared to the default background run ($x_b$), higher correlations and lower differences (RMSE, MAD, MRD, NMB) are observed. At the hourly observation sites, the comparison improved the most ($x_b$: {r = 0.73, RMSE = 3.67, MAD = 2.67, MRD = -0.22, NMB = -0.27, slope = 0.84} versus $x_{b,CrIS}$:{r = 0.82, RMSE = 2.98, MAD = 2.24, MRD = -0.12, NMB = -0.20, slope = 0.88}). Compared to observations from the passive sampler and wet-only sampler networks, the modelled monthly $NH_3$ surface concentration and $NH_4^+$ wet deposition fields now generally lie around the one-on-one line during spring

(March, April, May). There is, on the other hand, an overestimation in July and August now. Moreover, as a result of the decrease in CrIS-based $NH_3$ time factors to zero during winter, the $NH_3$ surface concentration and $NH_4^+$ wet deposition in December is underestimated in the $x_{b,CrIS}$ run.

Compared to the background runs ($x_b$ and $x_{b,CrIS}$), the two analysis runs ($x_a$ and $x_{a,CrIS}$) show higher correlations with all types

of measurements. The differences (RMSE, MAD, MRD, NMB) between the observed and modelled monthly $NH_3$ surface concentrations at the passive sampler sites are now, on the other hand, larger in the two analysis runs ($x_a$ and $x_{a,CrIS}$), illustrating an overall overestimation of the $NH_3$ concentrations in background regions. Although a large shift in the temporal distribution of the monthly $NH_4^+$ wet deposition is observed, the differences between the observed and modelled values remained similar. At the hourly observation sites, the comparison improved the most in the analysis run with the CrIS-based $NH_3$ time factors

($x_{a,CrIS}$). Here, compared to the default background run ($x_b$), higher correlations and smaller differences were found ($x_b$: {r = 0.73, RMSE = 3.67, MAD = 2.67, MRD = -0.22, NMB = -0.27, slope = 0.84} versus $x_{a,CrIS}$:{r = 0.83, RMSE = 2.83, MAD = 2.21, MRD = 0.03, NMB = -0.07, slope = 1.0}).





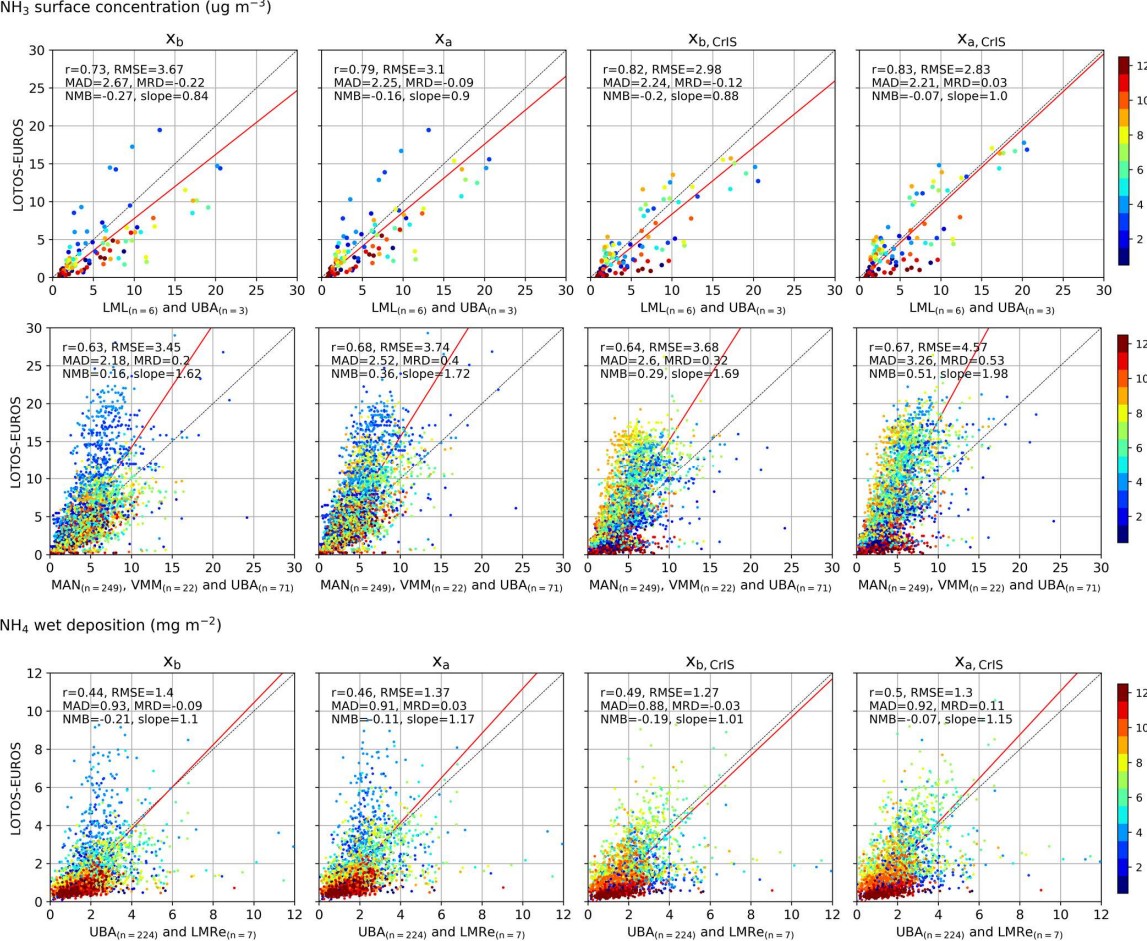

**Figure 13: Comparison of the modelled monthly mean NH₃ surface concentrations and NH₄⁺ wet deposition fields to in-situ observations. The colors indicate the month.**

### 3.4.3. Regional patterns

The modelled $NH_3$ surface concentrations were compared to observations from each passive sampler network separately. Fig. S15, S16 and S17 show comparison with the MAN network in the Netherlands, the UBA network in Germany and the VMM network in Belgium, respectively. In the default background run ($x_b$), the Dutch sites with relatively higher $NH_3$ surface concentrations are overestimated, most of which are located along the eastern border of the Netherlands. The highest correlation coefficients and lowest differences (RMSE, MAD) are found at the VMM network in Belgium. Here, the lower



$NH_3$ surface concentration sites are overestimated and the higher $NH_3$ concentrations sites are underestimated in the default background run ($x_b$). At the German UBA stations, the comparison lies more around the one-on-one line. The mean $NH_3$ surface concentrations at the sites close to the western border of Germany are generally overestimated in the default background run ($x_b$).

The use of the CrIS-based $NH_3$ time factors ($x_{b,CrIS}$) led to an overall increase in modelled mean $NH_3$ surface concentrations compared to the default background run ($x_b$). This led to a slight, overall increase in differences (RMSE and MAD) at all networks. Furthermore, steeper slopes were found at all three networks, i.e., the modelled $NH_3$ surface concentrations increased relatively more at sites with already higher concentrations. The same is observed in the two analysis runs ($x_a$ and $x_{a,CrIS}$), but to a greater extend. Compared to background runs ($x_b$ and $x_{b,CrIS}$), the differences (RMSE, MAD) between the modelled and observed concentrations were relatively higher at all networks. At the Dutch MAN network, a slightly higher correlation coefficient is observed.

Fig. S18 of the supplementary materials shows another comparison of the modelled and observed $NH_3$ surface concentrations at the hourly observation stations at daily resolution. Here, the correlation coefficient, root-mean-squared error RMSE, the mean difference MD and the slope are shown per site. The stations are located in different $NH_3$ emission regimes and are sorted by increasing $NH_3$ surface concentrations. The modelled $NH_3$ surface concentrations in the default background run ($x_b$) are generally overestimated at stations with low $NH_3$ emission regimes and underestimated at stations with medium to high $NH_3$ emission regimes. The use of the CrIS-based $NH_3$ time factors ($x_{b,CrIS}$) led to an improved comparison (higher correlation coefficient and lower RMSE) at the Dutch stations, but a worse comparison at the German stations. On a monthly basis, the comparison to the German UBA sites slightly worsened after the use of the CrIS-based $NH_3$ time factors ($x_{b,CrIS}$) (Fig. S19). The modelled $NH_3$ surface concentrations in the background run with the CrIS-based $NH_3$ time factors ($x_{b,CrIS}$) were, on the other hand, closer to the observations of the Dutch LML network in most months, with a lower differences (RMSE, MD) and slopes closer to 1. Here, the largest increase in correlation coefficients were found in March and April. In both analysis runs ($x_a$ and $x_{a,CrIS}$), the correlation coefficient improved and lower model-observation differences were found at all sites. Here, no clear distinction between sites located in different $NH_3$ emission regimes can be seen.

Compared to the default background run ($x_b$), the modelled $NH_3$ surface concentrations in the background run with the CrIS-based $NH_3$ time factors ($x_{b,CrIS}$) thus improved the most at Dutch stations located in medium to high $NH_3$ emission regimes. Most of the Dutch stations are located in the proximity of agricultural hotspots. The German stations, on the other hand, are located in background areas in central Germany, further away from major agricultural hotspots for $NH_3$. Fig. S8 of the supplementary materials shows the fitted CrIS-based $NH_3$ time factors at each site. The fitted $NH_3$ time factors at the majority of the Dutch stations show clear, identifiable peaks, in particular the spring peak. Moreover, most Dutch sites show clear year-to-year variations. For the German stations, on the other hand, the fitted $NH_3$ time factors are much flatter and show much less





interannual variation. This indicates that the observed CrIS-NH$_3$ surface concentrations at these locations remained around the same level, and thus that no clear (inter)annual patterns were present in the CrIS data.

In the Netherlands, the CrIS-based NH$_3$ time factors led to an improvement in the representation of the NH$_3$ spring peak. A time-series of the observed daily NH$_3$ surface concentrations at LML sites Valthermond and Zegveld are plotted in Fig. S20 of the supplementary materials. The modelled NH$_3$ surface concentrations in the default background run (x$_b$) start to rise too early in the year, particularly in 2014. In the background run with the CrIS-based NH$_3$ time factors (x$_{b,CrIS}$), both the start and the duration of the spring peak in NH$_3$ concentration improve. Here, the onset of the spring peak is delayed, better matching

the observed NH$_3$ timeseries.

### 4.1. Summary

In this study, the CrIS-NH$_3$ product is integrated into the LOTOS-EUROS chemical transport model using two different methods. In the first method, the CrIS-NH$_3$ surface concentrations were used to fit spatially varying NH$_3$ time factors to redistribute the NH$_3$ emission inputs in LOTOS-EUROS over the year. In the second method, the CrIS-NH$_3$ columns were

assimilated to adjust NH$_3$ emissions through local Ensemble Transform Kalman filtering in a top-down approach.

The fitted NH$_3$ time factors based on the CrIS-NH$_3$ surface concentrations led to a major temporal redistribution of the NH$_3$ emissions. In most regions, the updated NH$_3$ time profiles became flatter, with an overall decrease in spring (March to May) NH$_3$ emissions and an increase in NH$_3$ emissions in June to October. As a result, the mean modelled NH$_3$ fields between 2014

and 2018 spatially changed by up to +25% in NH$_3$ surface concentrations, -5 to +5% in NH$_3$ total column concentrations and -5 to +5% in NH$_x$ budget. The CrIS-based NH$_3$ time factors added an extra interannual variation of up to ±25% in the annual mean NH$_3$ concentrations and deposition fields. Data assimilation of the CrIS-NH$_3$ columns with the LETKF led to a unanimous increase in total NH$_3$ emissions. The modelled NH$_3$ fields between 2014 and 2018 changed with up to +30% in NH$_3$ surface concentrations, up to +20% in NH$_3$ total column concentrations and +10 to +25% in NH$_x$ budget. The largest

increases in NH$_3$ emissions (+30%) were found over the south of the Netherlands (Brabant), the west of Belgium (West-Vlaanderen) and a large region in northeastern Germany. The temporal distribution of the NH$_3$ emissions wasn't largely adjusted by the LETKF. The largest positive NH$_3$ emission updates took place in late summer and the beginning of autumn (July to September) and both increases and decreases in NH$_3$ emissions were observed in spring (March to May).

The modelled NH$_3$ surface concentration and NH$_4^+$ deposition fields were compared to in-situ observations. Our results illustrate that the strength of the first method, i.e., CrIS-based NH$_3$ time factors, primarily lies in improving the temporal distribution of the NH$_3$ emissions. Compared to in-situ networks, an overall increase in correlation coefficient and clear decrease in differences (RMSE, MAD, MRD, NMB) at the hourly observation and the wet-only sampler sites was observed. Moreover, time-series of observed daily NH$_3$ surface concentrations illustrate that using the CrIS-based NH$_3$ time factors





resulted in a better representation of both the onset and duration of the spring $NH_3$ peak in the Netherlands. The second method, data assimilation of the CrIS-$NH_3$ columns with the LETKF, improved the comparability to in-situ observation both spatially and temporally. Here, higher correlations with both annual and monthly observed mean $NH_3$ surface concentrations and $NH_4^+$ wet deposition were observed. This method also led to a decrease in differences (RMSE, MAD, MRD, NMB) at the hourly observation and the wet-only sampler sites. The mean $NH_3$ surface concentrations at the passive sampler sites, on the other

hand, were more strongly overestimated in both methods. The comparison to in-situ observations improved the most, both spatially and temporally, in the run where both methods are combined ($x_{a,CrIS}$). This illustrates that an initial, observation-based, rescaling of the $NH_3$ emissions enhances the adaptability of the LETKF, herewith thus improving the modelled $NH_3$ surface concentration and $NH_4^+$ wet deposition fields.

### 4.2. Discussion

#### 4.2.1. CrIS-based $NH_3$ time factors

The temporal redistribution of the $NH_3$ emissions after using the fitted CrIS-based $NH_3$ time factors led to a significantly better representation of the temporal variation in $NH_3$ emissions, especially the spring peak. Compared to in-situ observations, however, the $NH_3$ surface concentrations were overestimated in late summer and autumn (August to October). Further fine-tuning of the fitting algorithm could help to reduce the current overestimation and potentially improve the fitted $NH_3$ time

factors. For example, data filtering and selection criteria could be adapted. Narrowing the selection radius used for selecting the CrIS-$NH_3$ observations could for instance lead to a better representation of the $NH_3$ concentrations locally. This, however, may not always be possible, as a minimum number of observations is needed for a converging fit. Furthermore, the fitting algorithm currently doesn't allow for $NH_3$ area emissions during winter, because of the limited number of available CrIS observations at this time. As a result, the fitted $NH_3$ time factors show a relatively steep increase at the beginning of spring and

a decrease at the beginning of winter. This could lead to step-like functions in areas where clear $NH_3$ peaks in the CrIS-$NH_3$ data are absent. However, as this mainly occurs in areas with little to no $NH_3$ emissions, this shouldn't severely impact the modelled $NH_3$ concentrations in this study.

#### 4.2.2. Local Ensemble Transform Kalman Filter

The $NH_3$ emission updates computed by the Local Ensemble Transform Kalman Filter (LETKF) always remain tied to the

initial model fields by a certain uncertainty range. As such, data assimilation of the CrIS-$NH_3$ columns with the LETKF is mainly suitable for fine-tuning $NH_3$ emissions in regions where the $NH_3$ emissions are already relatively well known. The chosen LETKF configuration is for instance not able to correct for missing $NH_3$ emissions in areas where little or no initial $NH_3$ emissions are present. Furthermore, the LETKF is unable to resolve temporal patterns well without sensible input, as was illustrated in an experiment with homogeneous $NH_3$ emission fields (supplement section S1).






The LETKF filter settings used in this modelling setup ($\rho$ = 15 km, $\sigma$ = 0.5, $\tau$ = 3 days) led to a maximum emission increase of roughly ~30% on the initial $NH_3$ emissions for long-term simulations. The choice of these filter settings affects the adaptability of the LETKF, i.e., the achievable emission adjustments by correction factors. In this study, a temporal length scale $\tau$ of 3 days was chosen as a compromise between short time scales needed for irregular emissions (e.g., fertilizer

application) and longer time scales needed for regular emissions (e.g., stables and other point sources). Moreover, it matches the average satellite revisiting time per grid cell given the number of CrIS-$NH_3$ observations left after data selection (Fig. S21). A spatial correlation of $\rho$ = 15 km was chosen because it matches the footprint size of the satellite. Furthermore, as shown in section S1 in the supplement, increasing standard deviation $\sigma$ leads to larger, positive $\beta$ factors. To prevent further overestimations in background regions, a $\sigma$ of 0.5 was used for this region.


The current LETKF settings could for instance be improved by using spatially varying $\tau$ values. The choice of $\tau$ could be optimized for each emission category in the model. Locations with fertilizer application as dominant $NH_3$ emission source could for instance be set to lower $\tau$ values than locations with predominantly regular $NH_3$ sources. Another way to optimize the filter settings would be to study timeseries of the model-satellite discrepancies in more detail and base the choice of $\tau$ on

this. A more apparent memory effect (i.e., higher $\tau$) would be useful in areas with consistent model-satellite discrepancies, whereas in areas with incidental differences a lower $\tau$ would be more appropriate. Similarly, statistical analysis of the already computed emission perturbation factors $\beta$ could be performed. In this study, the model uncertainty follows a normal distribution in the current model setup. The distribution of the $NH_3$ concentrations, however, is, in reality, better approximated by a log-normal distribution. It would therefore be more realistic to use a log-normal distribution for the model uncertainty as

well. This would incidentally allow for larger correction factors when high $NH_3$ peaks are observed, for instance after fertilizer application.

In the current LETKF model setup, only the $NH_3$ emissions are perturbed. Thus, the discrepancies between the observed and modelled $NH_3$ concentrations are currently thus fully assigned to errors in the underlying model $NH_3$ emissions. However,

other model uncertainties could also cause these discrepancies, for instance uncertainties in other model inputs (e.g., other trace gas emissions) or parameterizations (e.g., deposition routines). In a follow-up study, it would be interesting to further investigate to the effect of an inverted LETKF setup, where model sink terms are perturbed instead of the source terms. However, the current setup is the most obvious as the $NH_3$ emissions are likely the largest source of model uncertainty in this area. It would also be interesting to assimilate in-situ observations and/or other satellite products (for instance IASI-$NH_3$)

simultaneously in a follow-up study.

### 4.2.3. Data products

Direct comparison of the observed and simulated $NH_3$ columns showed distinctly lower $NH_3$ total column concentrations in LOTOS-EUROS. This discrepancy could be the result of a systematic underestimation of the input $NH_3$ emission in LOTOS-



EUROS, or other model uncertainties. It could, on the other hand, also be partially related to the CrIS observations themselves.
Here, only CrIS observations with the highest quality flag (QF=5) were used, which for instance could have resulted in a bias towards observations with higher $NH_3$ concentrations or during good weather (e.g., no clouds), as these observations usually have a lower uncertainty. Moreover, an offset of approximately $\sim0.5 \times 10^{16}$ molecules/cm² is observed. This seems to be the effect of the detection limit of the CrIS instrument, which is unable to detect very low $NH_3$ concentrations. Furthermore, this, too, could be enhanced by the relatively strict data selection criteria used in this study, which favors higher $NH_3$ concentrations

that usually have a lower uncertainty. In the next version of the CrIS-$NH_3$ product, which was not yet available for this study, these non-detects are addressed. This might lead to lower $NH_3$ concentrations in background regions and partially solve this discrepancy. Moreover, this could also result in a better comparison with observations of the passive sampler networks.

The differences between the modelled and observed $NH_3$ concentrations and $NH_4^+$ wet deposition fields are partially related

to limitations in the spatial representativeness of the in-situ observations. The comparison of the different model runs to in-situ observations showed an overall overestimation at the passive sampler sites. These sites are mainly located in nature areas and therefore assumed to be representative of background regions with little to no $NH_3$ emissions. However, especially in the Netherlands, the landscape layout is very heterogenous and the nature areas are relatively small. As a result, at the current model grid size, each model pixel is likely to include other landscape elements than nature alone. The larger model scale

averages out the small-scale effects, thus leading to an overestimation. The opposite is true for the hourly observation sites located in medium to high $NH_3$ emission regimes. Especially at sites close to $NH_3$ emission sources, an underestimation is expected.

### 4.2.4. Conclusions

To conclude, satellite observed CrIS-$NH_3$ timeseries are helpful in improving $NH_3$ emissions, both spatially and temporally.

Our results illustrated that CrIS-$NH_3$ surface concentrations can be successfully used to fit spatially variable $NH_3$ time factors, which allows us to improve temporal $NH_3$ emission distributions relatively easy in a forward modelling setup. Comparison with in-situ $NH_3$ surface concentrations and $NH_4^+$ wet deposition observations showed that the fitted CrIS-based $NH_3$ time factors were particularly useful for adjusting the timing and duration of the $NH_3$ spring peak in medium to high $NH_3$ regimes. Moreover, the comparison showed that the CrIS-based $NH_3$ time factors improve the temporal distribution of the modelled

$NH_3$ surface concentrations and $NH_4^+$ wet deposition fields. Our results show that data assimilation of the CrIS-$NH_3$ columns data with the Local Ensemble Transform Kalman Filter (LETKF) improves the comparability with in-situ observations both spatially and, to a lesser extent, temporally, too. As the adaptability of the LETKF is limited by the uncertainty in the modelled fields, the strength of this method primarily lies in fine-tuning pre-existing $NH_3$ emission patterns. As a result, this method proved particularly useful for improving spatial $NH_3$ distributions in long-term simulations. Moreover, our results illustrated

that combining both methods enhanced the adaptability of the LETKF, and led to the largest improvements in modelled $NH_3$ surface concentration and $NH_4^+$ wet deposition fields compared to in-situ observations.



*Author contributions.* SvdG worked on the observation-based NH$_3$ time factors. The CrIS-NH$_3$ dataset was provided by ED and MWS. SvdG, ED, AS and RK worked on the modelling and data assimilation. JWE, ED, AS, MWS, RK and MS helped
with the interpretation of the results. SvdG wrote the paper with contributions from all co-authors.

*Competing interests.* The authors declare that they have no conflict of interest.

*Acknowledgements.* The authors would like to thank the Rijksinstituut voor Volksgezondheid en Milieu (RIVM), the
Vlaamse Milieumaatschappij (VVM) and the Umweltbundesambt (UBA) for providing observations of the NH$_3$ surface
concentrations and NH$_4^+$ wet deposition.

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
