# Peer review of "Data assimilation of CrIS-NH3 satellite observations for improving spatiotemporal NH3 distributions in LOTOS-EUROS."

_Atmospheric Chemistry and Physics, 2021_

## Referee Comment (RC2)

**Review of : "Data assimilation of CrIS-NH3 satellite observations for improving spatiotemporal NH3 distributions in LOTOS-EUROS"**

Accurate NH3 emissions and validation data sets are important inputs for modelers seeking to understand seasonal and spatial variability and long term trends in NH3 concentrations. While this holds true for many trace gases it is especially the case for NH3; NH3 varies rapidly in time and space, and with the exception of western Europe, there are few in situ networks. This data gap can be filled to some extent with satellite data, which can provide global coverage, albeit at a fairly coarse resolution (on the order of 15 km is typical for infrared sensors), for extended periods. This paper seeks to demonstrate and validate a method for improving NH3 estimates from LOTOS-EUROS using the retrieved NH3 profiles from the CrIS sensor on the SNPP platform. The method uses CrIS surface values to adjust the day-to-day variability in the emissions, and assimilates CrIS profiles using Local Ensemble Transform Kalman Filter approach. The model output before and after the updates, is evaluated against in situ data from various networks in the Netherlands, Belgium and Germany, which provide a mix of observations from passive samplers, miniDOAS and wet-only samplers, with varying temporal resolution, from hourly to monthly.

The paper provides an excellent picture of NH3 emissions, surface concentrations and total columns, as well as NHx deposition in northwestern Europe as modelled by LOTOS-EUROS: both magnitudes and variability are well characterized. The use of the CrIS data to attempt to improve model performance is in general well explained and the validation results are interesting. The shift in the emissions peak from spring to later summer/fall is notable and improves the agreement with the in situ data in the Netherlands.

The one disappointing conclusion of the paper is that this method works best for regions where emissions are fairly well known; in these areas it can provide useful adjustments. Otherwise it appears to have little application.

The paper should be published after the minor revisions suggested below have been made.

**Medium comments**

**Comment 1:** The authors should explain how the temporal variability is calculated in the background runs and why improving it is important; it took a few readings to understand that it is not the hourly profile that is being changed.

**Minor edits and comments**

**Line 18:** column data can be interpreted as total column; maybe use profile instead here and throughout?

Line 63: spatial and temporal distribution…

Line 73: Need to check if NH3 products are being generated from IASI-NG

Line 74: CrIS has greater sensitivity to near-surface…

Line 76: These trace gas measurements have opened up…

Line 89: for the daily distribution …

Line 91: Using a Local …

Line 92: approach as a data-assimilation system, which enhances existing spatial patterns. (can the authors explain why?)

Line 99: that can …

Line 124: has greater sensitivity to NH3 close to the surface due to its low spectral noise  … and its 1:30 pm observation time, which coincides with the time of day with highest thermal contrast.

Line 161:  Gaussian curve  to scale NH3 surface concentrations(see section 2.3.3) from CrIS in each cell.

Line 175: that are mainly dependent…

Line 186: to adjust the daily variability in the hourly profiles

Line 186: Explain what qflag=3 entails and why it was chosen ( to increase the number of available points?)

Line 194: differs by region…

Line 195: and chemical conditions.

Line 196: The factor is derived  from …

Line 200: Colors in Figure S2 are not defined

Line 201: Explain a bit why it is important to avoid flattening the spring peak

Line 232: What does linearization of h(x) to x mean here?

Line 235: The ~N(0,R) is not familiar to all readers

Line 240:

Line 241: Define G

Line 246: The connection between eq. 8 and eq. 9 is not well made

Line 264: Explain why QF=5 is being used here.

Line 310: the time factors alone

Line 367: Similarly

The statistics in Figures 12 and 13 should be presented also in a table, with the most important results highlighted.

Also need a table here showing the type of instruments in each location.

Line 625: In this study….. This is a good point.

---

## Author Comment (AC1)

Authors response

The authors would like to thank referee #1 and #2 for their valuable comments. We have changed the manuscript accordingly. The changes made to the manuscript are marked in  (deleted text) or green (added or changed text/figures) in the attached word file.

On behalf of all authors,
Shelley van der Graaf

Response to anonymous Referee #1

Dear Referee #1,

We would like to thank you for taking the time to review our manuscript and your encouraging comments!

Kind regards,
Shelley van der Graaf

Response to anonymous Referee #2

Dear Referee #2,

Thank you for your valuable comments! We have written responses to your notes individually in blue in the text below. Thank you again for taking the time to review our manuscript.

Kind regards,
Shelley van der Graaf

Medium comments

Comment 1: The authors should explain how the temporal variability is calculated in the background runs and why improving it is important; it took a few readings to understand that it is not the hourly profile that is being changed.

Minor edits and comments

Line 18: column data can be interpreted as total column; maybe use profile instead here and throughout? We have changed this to "the CrIS-NH$_3$ profile"
Line 63: spatial and temporal distribution... we changed this
Line 73: Need to check if NH3 products are being generated from IASI-NG
Line 74: CrIS has greater sensitivity to near-surface... changed
Line 76: These trace gas measurements have opened up... changed
Line 89: for the daily distribution ... changed
Line 91: Using a Local ... changed
Line 92: approach as a data-assimilation system, which enhances existing spatial patterns. (can the authors explain why?) We removed this line here because it is explained later in the manuscript.
Line 99: that can ... changed
Line 124: has greater sensitivity to NH3 close to the surface due to its low spectral noise ... and its 1:30 pm observation time, which coincides with the time of day with highest thermal contrast. We have added this to the text.
Line 161: Gaussian curve to scale NH3 surface concentrations (see section 2.3.3) from CrIS in each cell. We changed this.
Line 175: that are mainly dependent... changed
Line 186: to adjust the daily variability in the hourly profiles changed
Line 186: Explain what qflag=3 entails and why it was chosen (to increase the number of available points?) This is correct, we decided to use the observations with a quality-flag of 3 or higher because the number of observations with quality flag 5 per grid cell was insufficient for a decent fit.
Line 194: differs by region... changed

Line 195: and chemical conditions. We changed this

Line 196: The factor is derived from ... changed

Line 200: Colors in Figure S2 are not defined We have added a color bar to the figure.

Line 201: Explain a bit why it is important to avoid flattening the spring peak We elaborated more on this and added an explanation.

Line 232: What does linearization of h(x) to x mean here? We describe this later in the next paragraph (2.4.4), we added a reference to this paragraph in the text.

Line 235: The ~N(0,R) is not familiar to all readers The error is taken from a normal distribution with mean 0 and standard deviation R, we added some lines to the text explaining this.

Line 241: Define G Matrix G is the gridding operator, used to "regrid" the model and satellite layers. We added some lines to the text explaining this.

Line 246: The connection between eq. 8 and eq. 9 is not well made The missing link between the two equations, $y^{true} = h(x^{true}) + v$, is added to the paragraph.

Line 264: Explain why QF=5 is being used here. As the data-assimilation system navigates between the model and observational uncertainty, we decided to use only observations with a low uncertainty (QF=5). Observations with a high uncertainty are far more likely to lead to small or no emission updates in the analysis runs, and therefore their impact is more limited.

Line 310: the time factors alone changed

Line 367: Similarly changed

The statistics in Figures 12 and 13 should be presented also in a table, with the most important results highlighted. Also need a table here showing the type of instruments in each location. We have added the requested table with a summary of the computed statistics to the manuscript (Table 2).

Line 625: In this study..... This is a good point.

---

## Author Response (AR2)

Authors response
The authors would like to thank referee #1 and #2 for their valuable comments. We have changed the manuscript accordingly. The changes made to the manuscript are marked in  (deleted text) or green (added or changed text/figures) in the attached word file.

On behalf of all authors,
Shelley van der Graaf

Response to anonymous Referee #1
Dear Referee #1,

We would like to thank you for taking the time to review our manuscript and your encouraging comments!

Kind regards,
Shelley van der Graaf

Response to anonymous Referee #2
Dear Referee #2,

Thank you for your valuable comments! We have written responses to your notes individually in blue in the text below. Thank you again for taking the time to review our manuscript.

Kind regards,
Shelley van der Graaf

Accurate $NH_3$ emissions and validation data sets are important inputs for modelers seeking to understand seasonal and spatial variability and long-term trends in $NH_3$ concentrations. While this holds true for many trace gases it is especially the case for $NH_3$; $NH_3$ varies rapidly in time and space, and with the exception of western Europe, there are few in situ networks. This data gap can be filled to some extent with satellite data, which can provide global coverage, albeit at a fairly coarse resolution (on the order of 15 km is typical for infrared sensors), for extended periods. This paper seeks to demonstrate and validate a method for improving $NH_3$ estimates from LOTOS-EUROS using the retrieved $NH_3$ profiles from the CrIS sensor on the SNPP platform. The method uses CrIS surface values to adjust the day-to-day variability in the emissions, and assimilates CrIS profiles using Local Ensemble Transform Kalman Filter approach. The model output before and after the updates, is evaluated against in situ data from various networks in the Netherlands, Belgium and Germany, which provide a mix of observations from passive samplers, miniDOAS and wet-only samplers, with varying temporal resolution, from hourly to monthly. The paper provides an excellent picture of NH3 emissions, surface concentrations and total columns, as well as NHx deposition in northwestern Europe as modelled by LOTOS-EUROS: both magnitudes and variability are well characterized. The use of the CrIS data to attempt to improve model performance is in general well explained and the validation results are interesting. The shift in the emissions peak from spring to later summer/fall is notable and improves the agreement with the in situ data in the Netherlands.

The one disappointing conclusion of the paper is that this method works best for regions where emissions are fairly well known; in these areas it can provide useful adjustments. Otherwise, it appears to have little application.
The LETKF indeed seems to work best in regions where the emissions are fairly well known. The computed emission adjustments remain tied to the model input emissions, also with different filter settings, this is intrinsic to the method. As such, the applicability of the LETKF system by itself is indeed limited when emission estimates are largely incorrect in terms of their spatial distribution or when sources are missing. However, as we have illustrated when combining the two methods described in our manuscript, much greater emission updates can be achieved when the $NH_3$ emissions are initially scaled. In this manuscript, we focused on the temporal distribution of modelled $NH_3$ emissions, however, a comparable two-step approach is thinkable when it comes to spatial distribution. For instance, the locations of missing emission sources could first be (roughly) estimated from $NH_3$ satellite observations, before applying the LETKF. Using such a hybrid approach, the LETKF could also be applicable in regions where the emissions are less known.

Medium comments

Comment 1: The authors should explain how the temporal variability is calculated in the
background runs and why improving it is important; it took a few readings to understand that it is
not the hourly profile that is being changed. In the background runs, fixed time profile per emission category are
used. For virtually all agricultural emission sources (except point-sources, for instance stables), this is the MACC-III
profile that is used as an initial guess for the fitting of the $NH_3$ time factors. We made some adjustment to the text to
explain this more clearly.

Minor edits and comments

Line 18: column data can be interpreted as total column; maybe use profile instead here and throughout? We have
changed this to "the CrIS-$NH_3$ profile"
Line 63: spatial and temporal distribution... we changed this
Line 73: Need to check if NH3 products are being generated from IASI-NG
Line 74: CrIS has greater sensitivity to near-surface... changed
Line 76: These trace gas measurements have opened up... changed
Line 89: for the daily distribution ... changed
Line 91: Using a Local ... changed
Line 92: approach as a data-assimilation system, which enhances existing spatial patterns. (can
the authors explain why?) We removed this line here because it is explained later in the manuscript.
Line 99: that can ... changed
Line 124: has greater sensitivity to NH3 close to the surface due to its low spectral noise ... and
its 1:30 pm observation time, which coincides with the time of day with highest thermal contrast. We have added
this to the text.
Line 161: Gaussian curve to scale NH3 surface concentrations (see section 2.3.3) from CrIS in
each cell. We changed this.
Line 175: that are mainly dependent... changed
Line 186: to adjust the daily variability in the hourly profiles changed
Line 186: Explain what qflag=3 entails and why it was chosen (to increase the number of
available points?) This is correct, we decided to use the observations with a quality-flag of 3 or higher because the
number of observations with quality flag 5 per grid cell was insufficient for a decent fit.
Line 194: differs by region... changed
Line 195: and chemical conditions. We changed this
Line 196: The factor is derived from ... changed
Line 200: Colors in Figure S2 are not defined We have added a color bar to the figure.
Line 201: Explain a bit why it is important to avoid flattening the spring peak We elaborated more on this and added
an explanation.
Line 232: What does linearization of h(x) to x mean here? We describe this later in the next paragraph (2.4.4), we
added a reference to this paragraph in the text.
Line 235: The ~N(0,R) is not familiar to all readers The error is taken from a normal distribution with mean 0 and
standard deviation R, we added some lines to the text explaining this.
Line 241: Define G Matrix G is the gridding operator, used to "regrid" the model and satellite layers. We added
some lines to the text explaining this.
Line 246: The connection between eq. 8 and eq. 9 is not well made The missing link between the two equations,
$y^{true} = h(x^{true}) + v$, is added to the paragraph.
Line 264: Explain why QF=5 is being used here. As the data-assimilation system navigates between the model and
observational uncertainty, we decided to use only observations with a low uncertainty (QF=5). Observations with a
high uncertainty are far more likely to lead to small or no emission updates in the analysis runs, and therefore their
impact is more limited.
Line 310: the time factors alone changed
Line 367: Similarly changed
The statistics in Figures 12 and 13 should be presented also in a table, with the most important
results highlighted. Also need a table here showing the type of instruments in each location. We have added the
requested table with a summary of the computed statistics to the manuscript (Table 2).

Line 625: In this study..... This is a good point.